# Learning Replacement Variables in interpretable rule-based Models

**Lena Stempfle** [1] , **Fredrik D. Johansson** [1]

## Abstract

Rule models are favored in many prediction tasks due to their interpretation using natural language and their simple presentation. When learned from data, they can provide high predictive performance, on par with more complex models. However, in the presence of incomplete input data during test time, standard rule models' predictions are undefined or ambiguous. In this work, we consider learning compact yet accurate rule models with missing values at both training and test time, based on the notion of replacement variables. We propose a method called `MINTY` which learns rules in the form of disjunctions between variables that act as replacements for each other when one or more is missing. This results in a sparse linear rule model that naturally allows a trade-off between interpretability and goodness of fit while being sensitive to missing values at test time. We demonstrate the concept of `MINTY` in preliminary experiments and compare the predictive performance to baselines with potential applications in clinical scoring systems.

## 1. Introduction

Rule-based models, such as risk scores, rule lists, and linear rule models, are favored in prediction problems and domains where interpretability is a concern (Fürnkranz et al., 2012; Wei et al., 2019; Margot & Luta, 2021). For example, clinical scoring systems are defined using a small number of rules with associated points that add up to a score, indicating, e.g., the risk of mortality for a patient (Knaus et al., 1991). In the same domains, it is common for some variables used in rules to be unobserved at the time of prediction, due to varying tool availability, examination protocols, or heterogeneous data sources (Madden et al., 2016). Despite this, most rule-based models lack built-in principled ways for making predictions with missing values. Approaches to prediction with incomplete data, include imputation (Rubin, 1976), Bayesian modeling, fallback default rules (Chen & Guestrin, 2016), weighted estimating equations (Ibrahim et al., 2005) and prediction with missingness indicators (Le Morvan et al., 2020). Drawbacks of existing methods are that they are

specific to a non-interpretable model class or that they reduce the interpretability of rule-based models by relying on auxiliary models which may not be interpretable (imputation, estimation weighting) or on parameters associated with missingness itself (default rules, missingness indicators) (Stempfle & Johansson, 2022). If there is redundancy in the covariates set, where two variables have similar associations to the outcome, we may not need to observe both of them to predict accurately. Instead, redundant variables $A$ and $B$ could be used as replacements for each other when one of them is missing: "If $A$ is not available, use $B$", or "If $B$ is not available, use $A$".

Below we show an example of rules which illustrate how replacement variables can be used in the context of linear rule models with binary covariates; if at least one variable in each rule is observed and active, the prediction is the same whether other variables in the rule are missing.

$$\textbf{prediction} = \text{Coefficient}_1(\text{Variable}_1 \text{ OR Variable}_2)$$
$$+ \text{Coefficient}_2(\text{Variable}_3 \text{ OR Variable}_4)$$

Using replacement variables also avoids direct dependence on imputation or missingness indicators. In this project, we aim to *learn replacement variables for missing values at test time using a rule-base interpretable model*. Replacements for unobserved variables should be learned during the training phase and then retrieved at test time. We propose a new methodology, `MINTY` which utilizes replacement variables, defined by disjunctions of literals, in generalized linear rule models. Replacement variables indicate which features can be alternatively used in situations where the original feature was not measured. In addition, they ensure a comparable predictive power to their original counterpart.

## 2. Prediction with missing values at test time

We consider a supervised learning problem of predicting a continuous outcome of interest $Y \in \mathbb{R}$ based on a vector of $d$ input features $X = [X_1, ..., X_d]^\top$. In our setting, features may be missing both at training time and at test time, as indicated by a random *missingness mask* $M = [M_1, ..., M_d]^\top \in \{0, 1\}^d$ such that $M_j = 0$ if $X_j$ is observed and $M_j = 1$ otherwise. We let $\tilde{X} \in (\mathbb{R} \cup \{\texttt{NA}\})^d$ indicate the partially observed feature vector.

We are given a training data set of samples $(x_i, m_i, y_i)$ for

$i = 1, \ldots, n$, drawn i.i.d. from a distribution $p$, with $x_i = [x_{i1}, \ldots x_{id}]^\top$ the feature vector of sample $i$ with missing values, and $m_i, y_i$ defined analogously. For convenience, we let $\mathbf{X} \in (\mathbb{R} \cup \{\texttt{NA}\})^{n \times d}, \mathbf{M} \in \{0,1\}^{n \times d}, \mathbf{Y} \in \mathbb{R}^{1 \times d}$ denote data matrices of features, missingness masks and outcomes for all observations, respectively.

We assume that all features $X_j$ represent logical literals, taking values in $\{0, 1\}$, where $X_{ij} = 1$ represents that literal $j$ is true for observation $i$. In a health care example, feature $j$ may represent the literal $Age \geq 70$ and a patient $i$ that is 73 years old would have the observation $x_{ij} = 1$. Our goal is to predict $Y$ *under missingness* $M$ in $X$ using functions $f : (\mathbb{R} \cup \{\texttt{NA}\})^d \to \mathbb{R}$, with minimum risk with respect to the squared loss on $p$,

$$\min_f R(f), \text{ where } R(f) := \mathbb{E}_{\tilde{X}, Y \sim p}[(f(\tilde{X}) - Y)^2]. \quad (1)$$

We assume that features and their missingness have the same distribution at test time as during training. A common strategy for learning and prediction with missing values is to impute unobserved variables based on observed ones (Rubin, 1976) and proceed as if no values were missing in the first place. However, when missingness itself depends on unobserved values—variables are missing not-at-random (MNAR)—this strategy is generally suboptimal, (Jamshidian & Mata, 2007).

In our setting, under the assumption that $Y$ has centered, additive noise, $Y = g(X, M) + \epsilon$ where $\mathbb{E}[\epsilon] = 0$, the Bayes-optimal predictor of $Y$ is $f^* = \mathbb{E}[Y = 1 \mid \tilde{X}, M]$ (Morvan et al., 2021), depending directly on the missingness mask $M$ itself. However, when interpretability is wanted, letting models include features such as "Age is missing" may be undesirable.

## 3. Methodology

We propose MINTY, a linear rule model for learning replacement variables when values expected by the model may be unobserved at test time. Our goal is to obtain small, interpretable models with high predictive performance when inputs are incomplete. We first describe the model class and then show how we solve the regression task using constrained optimization. MINTY is a generalized linear rule model (Wei et al., 2019) with three main components:

1. *Rule definitions* $z_{\cdot k} \in \{0, 1\}^d$, for rules $k = 1, ..., K$, defining logical rules in terms of $d$ features (literals)

2. *Rule activations* $a_{ik} \in \{0, 1\}$, which indicate whether individual $i = 1, ..., n$ satisfies rule $k$

3. *Rule coefficients*, $\beta = [\beta_1, ..., \beta_K]^\top \in \mathbb{R}^K$, where $\beta_k$ relates rule $k$ to the predicted outcome. By letting rule 1 always be true, $\beta_1$ takes the role of an intercept.

MINTY handles missing values by making predictions based on rules formed as *disjunctions* of literals, such as "(Age > 20) or (Female)". If the value of "Age" is missing, the rule depends only on the value of "Female". If none of the features in a disjunction is observed, the rule is inactive—acting like zero-imputation. To prevent this from happening, at training time, we require that, for each observation and each rule, at least one literal is observed. We formalize the MINTY model as follows.

Given an observation $x_i$, with missingness, let $\bar{x}_i$ denote its zero- imputation, $\bar{x}_i = x_i \mathbb{1}[x_i \neq \texttt{NA}]$. Then, define the activation of rule $k$ for $x_i$ to be,

$$a_{ik} = \bigwedge_{j=1}^{d} z_{jk} \bar{x}_{ij} = \max_{j \in [d]} z_{jk} \bar{x}_{ij}$$

where $z_{jk} = 1$ indicates that literal (feature) $j$ is included in disjunction (rule) $k$. Given such activations, the prediction for an input $x_i$ is made as $\hat{y}_i = \sum_{k \in S} \beta_k^\top a_{ik}$, where $S$ denotes the set of disjunctions under consideration, defined by indicators $z_{jk}$. We aim to find both a set of rules $S$ and coefficients $\beta$ that minimize the regularized empirical risk,

$$\min_{\beta, S} \frac{1}{n} \sum_{i=1}^{n} (\sum_{k \in S} \beta_k a_{ik} - y_i)^2 + \sum_k \lambda_k |\beta_k|, \quad (2)$$

with an $\ell_1$-penalty $\lambda_k |\beta_k|$ for including rule $k$. By choosing $\lambda_k$, we can control the number and size of rules used by the model. When generating the rule set, we restrict $S$ to only include rules where at least one of the variables in each rule $k$ is measured for each subject $i$.

### 3.1. Optimization

By letting $S$ be the set of all possible disjunctions $\mathcal{K} = \{0, 1\}^d$, our learning problem (2) reduces to a LASSO problem with known solvers, but with a number of rules and coefficients growing exponentially in $d$. Even for moderate-size problems, it would be intractable to enumerate all of them. Instead, we follow the column-generation strategy by Wei et al. (2019), which intelligently searches the space of disjunctions and builds up $S \subseteq \mathcal{K}$ incrementally. The idea is to first solve a restricted problem with a small set of candidate rules $S_0$, in our case just the intercept rule, and then iterative adding new candidates based on the optimal dual solution of the restricted problem. A rule is selected based on the marginal benefit (or partial derivative) of introducing it to the restricted problem. If the partial derivative for the most promising column is non-negative, the procedure terminates. We modify their approach by requiring that each added rule has at least one observed feature for each observation in the training set. Given a current set of disjunctions $S$ and coefficients $\beta$, a new rule is added to $S$ by finding a disjunction that can explain the largest part of

the residual of the current model, $\mathbf{R} = \mathbf{A}\beta - \mathbf{Y}$, where $\mathbf{A} = [a_{1.}, \ldots, a_{n.}]^\top$ is the matrix of rule assignments for all observations $i = 1, \ldots, n$ in the training set. (Wei et al., 2019) show that such a rule $z$ may be found by solving the following optimization problem for both signs of the first term in the objective.

$$
\begin{aligned}
&\underset{\substack{a \in \{0,1\}^n \\ z \in \{0,1\}^d}}{\text{minimize}} \quad \pm \frac{1}{2n} \sum_{i=1}^n r_i a_i + \lambda_0 + \lambda_1 \sum_{j=1}^d z_j \\
&\text{subject to} \quad a_i = \sum_{k=1}^K \max(x_{ij} z_j) \\
&\qquad\qquad \forall i : \max_j (1 - M_{ij}) z_j \geq 1,
\end{aligned}
\tag{3}
$$

where $n_m$ is the number of samples. The first constraint in (3) makes sure that rule activations $a_{ik}$ correspond to a disjunction of literals $x_{ij}$ as indicated by $z_j$. For the second constraint, we require that, for all rules, at least one of the included literals $j : z_j = 1$ is observed for every individual $i$. To find an approximate solution to (2), we start with a subset $S_0$ of rules, solve (2) with respect to $\beta$ for this set, and compute the residual $\mathbf{R}$ for the current model. Then, repeatedly, a single rule is added to $S$ based on maximizing its correlation with the residual $\mathbf{R}$, solving (3), and the coefficients $\beta$ are refit. When no rule can be found with a negative solution to (3), the algorithm terminates and the coefficients $\beta$ are refit one last time.

**Relaxed Version of** $MINTY$  In $MINTY$ with zero imputation, the missingness constraint in (3) is always inactive, leading to a larger model class that has been proven to benefit learning disjunctions. As a compromise, instead of having the constraint that every rule be measured for every training sample, we minimize the number of rules where no variable is measured. We introduce a variation of the original algorithm, $\texttt{MINTY}_{relaxed}$ shown in (4) in the Appendix A.2.

## 4. Experiments

We evaluate the $\texttt{MINTY}$ model[1] on synthetic data aiming to answer two main questions: How does the accuracy of $\texttt{MINTY}$ compare to baseline models; How do replacement variables affect performance and interpretation?

**Experimental Setup**  In the column generation subproblem, to find values for $\beta$, given rule definitions $S$, we use the $\texttt{LASSO}$ implementation in scikit-learn (Buitinck et al., 2013) with covariate weighting to achieve variable-specific regularization strength. We iteratively add variables to $S$ by

optimizing (3) using Gurobi, a general-purpose optimization solver (Gurobi Optimization, LLC, 2023).

The objective function regularizes each rule $z_{.k}$ with strength $\lambda_k = \lambda_0 + \lambda_1 \|z_{.k}\|_0$, penalizing high numbers of unmeasured literals per rule. The values of $\lambda_0$ and $\lambda_1$ range within $[10e^{-3}, 0.1]$. In $\texttt{MINTY}_{relaxed}$ we range the regularization parameter $\gamma$ within $[10e^{-3}, 0.1]$ and normalized over all samples $n$.

**Baseline models**  The baselines we used included $\ell_1$-regularized linear regression models, also known as $\texttt{LASSO}$, a decision tree model ($\texttt{DT}$) (Pedregosa et al., 2011), and a GXBoost ($\texttt{XGB}$), where missing values are supported by default (Chen & Guestrin, 2016). We also implement the NeuMiss ($\texttt{NEUMISS}$) network, which approximates a specialization term on observed data (along with per-pattern biases) using a deep neural network where both covariates and missingness mask are given as input, sharing parameters across patterns (Le Morvan et al., 2020). These models were applied to imputed data using zero imputation ($I_0$), whereas, in future experiments, more sophisticated methods such as Multiple Imputation by Chained Equations (MICE) (Buitinck et al., 2013) can be used. Methods such as $\texttt{XGB}$, $\texttt{NEUMISS}$, and $\texttt{MINTY}$ do not rely on imputation. More details are given in Appendix A.3.

### 4.1. Simulated Data

We use simulated data to illustrate the process of generating replacement variables focusing on predictive performance and interpretablity. We sample $n \times d$ independent binary input features such that $X_{ij} \sim \text{Bernoulli}(p)$, with $p = 0.3$. We sample 2000 data points. The outcome $Y$ is given by a disjunctive linear rule model of $S$, without noise (see Table 2). We limit our experiments to the missing-at-random (MAR) mechanism given by (Mayer et al., 2019). The proportion of missing values to generate for variables that will have missing values is set to $p_{miss} = 0.2$. To give the rules meaningful descriptions we constructed column names based on the features available in the Alzheimer's Disease Neuroimaging Initiative (ADNI)[2] database.

### 4.2. Results

We report the predictive performance, as the mean square error (MSE) and the $R^2$ score of $\texttt{MINTY}$ compared to baselines over 10 draws (Table 1). The statistical uncertainty of the average error is measured with its square root, which is a standard deviation and expressed by $95\%$ confidence intervals over the test set.

For the regression task, $\texttt{MINTY}_{relaxed}$ and $\texttt{XGB}$ perform best with an $R^2$ of 0.73 and the same confidence intervals.

---

[1]The code to reproduce the experiments is available at `https://github.com/Healthy-AI/minty`

[2]`http://adni.loni.usc.edu`

*Table 1.* Results for the synthetic data comparing `MINTY` to the baseline models.

| MODEL | $R^2$ | $MSE$ |
|---|---|---|
| LASSO ($\text{I}_{zero}$) | 0.40 (0.32, 0.47) | 3.94 (3.67, 4.22) |
| XGB ($\text{I}_{zero}$) | **0.73 (0.68, 0.78)** | **1.73 (1.55, 1.92)** |
| DT ($\text{I}_{zero}$) | 0.68 (0.63, 0.73) | 2.08 (1.88, 2.28) |
| NEUMISS | 0.58 (0.51, 0.64) | 2.78 (2.55, 3.01) |
| $\text{MINTY}_{relaxed}$ | **0.73 (0.68, 0.78)** | **1.78 (1.60, 1.96)** |
| MINTY | 0.71 (0.66, 0.77) | 1.88 (1.70, 2.08) |

Validation performance resulted in selecting $\lambda_0 = 0.1$, and $\lambda = 0.1$ and a $\gamma$ value of 1 for $\text{MINTY}_{relaxed}$. `MINTY` achieves a slightly lower $R^2$ of 0.71 using $\lambda_0 = 0.1$, and $\lambda = 0.1$ than its algorithmic relaxer variation. When comparing `DT` ($R^2$ of 0.68) and `XGB` we can assume that `XGB` benefits from its ability to handle missing data without relying on imputation. However, the black-box nature of `XGB` and `NEUMISS` ($R^2$ of 0.58) is not conducive to reasoning about missingness to improve prediction comprehension. With an $R^2$ score of 0.40, `LASSO` seems to have a disadvantage due to its linear function class.

Results are shown in Figure 2 in the Appendix, comparing the $R^2$s with estimator-specific complexity measurements. We see that both `MINTY` models perform better than the baselines with a small number of non-zero coefficients that ensure lower model complexity. `XGB` achieves consistent performance across estimators, but could be difficult to interpret with a larger number of estimators (and an even larger number of parameters). In a `DT`, neighboring leaves are similar to each other as they share the path in the tree. As the number of leaves increases, variance in the performance increases and perhaps compromises interpretability. While `LASSO` is the simplest model, its performance is the lowest. In summary, both `MINTY` models perform comparably to the baselines, with quite tight confidence intervals.

**Customized rule sets** We present descriptions for individual instances and describe the rules relevant to them while the coefficients sum up to the prediction made by the model (Table 2). Variables that are not measured are removed from the rules, and the coefficients of the rules that become equal due to the removed variables are summed up. The simple representation supports domain experts, such as clinicians to make use of `MINTY` in their decision-making.

In Table 2, we compare a set of ground truth rules (top Table) to learned rules (bottom Table) from generated data. We interpret the results by saying that the first rule including (Sex= $female$) as a literal does not capture any rule in the true rule set. The literals in the third rule match those in the upper table, with the rounded coefficients matching perfectly. The literals in the last two rules in each table are correctly learned, however, the coefficients differ (+3 and

*Table 2.* Customized rule sets for predictions based on the ground true rule set $S$ (Top table). Learned rules set with corresponding coefficients in the bottom table are based on `MINTY`. The results are based on a generated data set with n = 2000 samples and a $p_{miss} = 0.2$

| TRUE RULES | COEFF. |
|---|---|
| (AGE$\geq$70) OR (SEX= $female$) | +2 |
| (HEART RATE$\geq$120BPM) OR (EDUCATION$_{low}$) | +3 |
| (EDUCATION$_{low}$) OR (PRIOR AD DIAGNOSIS) | +2 |
| (AGE$\geq$70) OR (HEART RATE$\geq$120BPM) | -5 |
| INTERCEPT | +0 |
| LEARNED RULES | COEFF. |
| (EDUCATION$_{low}$) OR (SEX= $female$) | +1.18 |
| (HEART RATE$\geq$120BPM OR (EDUCATION$_{low}$) | +1.69 |
| (EDUCATION$_{low}$) OR (PRIOR AD DIAGNOSIS) | +1.55 |
| (AGE$\geq$70) OR (HEART RATE$\geq$120BPM) | -2.97 |
| INTERCEPT | +0.36 |

+1.69 for (Education$_{low}$) OR (Prior AD diagnosis)). The coefficient for the last rule shows the right influence on the prediction but the learned coefficients should be higher in value to match the true coefficient (-5).

## 5. Related work

We focused on developing an interpretable rule-based model by learning variables that may act as replacements for each other. The literature on interpretable machine learning models contains other methods where learning replacements variables might be beneficial for prediction with test-time missingness. For example, risk scores provide a way to add interpretability in applications with domain-specific constraints (Ustun & Rudin, 2019). They comprise simple logistic-linear models but depend on imputation. The XGBoost (Chen & Guestrin, 2016) offers an alternative by assigning default paths for missing variables. The NeuMiss network proposes a new type of non-linearity: multiplication by the missingness indicator (Le Morvan et al., 2020).

## 6. Conclusion

We studied prediction with missingness where groups of variables are correlated but one or more may be missing at training and test time. We proposed the rule-based interpretable model `MINTY` for learning replacement variables for linear combinations of decision rules. Empirical results on synthesized data show that `MINTY` achieves comparable performance to baseline models. If meaningful variables are present, they can be evaluated by a domain expert for intuitiveness as replacements and allow for applications, e.g. in clinical scoring systems. In future work, real-world data and various missingness mechanisms can be studied.

## Software and Computing Infrastructure

The computations required resources of 4 compute nodes using two Intel Xeon Gold 6130 CPUS with 32 CPU cores and 384 GiB memory (RAM). Moreover, a local disk with the type and size of SSSD 240GB with a local disk, usable area for jobs including 210 GiB was used. Inital experiments are run on a Macbook using macOS Montery with a 2,6 GHz 6-Core Intel Core i7 processor.

## Acknowledgements

We thank the reviewers for their valuable feedback.

This work was partly supported by WASP (Wallenberg AI, Autonomous Systems and Software Program) funded by the Knut and Alice Wallenberg foundation.

The computations were enabled by resources provided by the Swedish National Infrastructure for Computing (SNIC) at Chalmers Centre for Computational Science and Engineering (C3SE) partially funded by the Swedish Research Council through grant agreement no. 2018-05973.

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

# A. Appendix

## A.1. Real world data sets

**ADNI** The compiled data set includes 1337 subjects that were preprocessed by one-hot encoding of the categorical features and dichotomized for the numeric features. The processed data has 76 features after the binning process. The regression task targets predicting the result of the cognitive test ADAS13 (Alzheimer's Disease Assessment Scale) at a 2-year follow-up (Mofrad et al., 2021) based on available data at baseline.

**MIMIC-III** The compiled data set includes 304 subjects that were preprocessed by one-hot encoding of the categorical features and dichotomized for the numeric features. The processed data has 164 features after the binning process.

## A.2. Modification to `MINTY`$_{relaxed}$

Note, $\delta$ is a binary relaxed variable indicating if at least one literal $j$ per rule $k$ for each individual is measured. We add a regularization term $\gamma > 0$ normalized of $n$ samples to indicate how much we emphasize individuals not fulfilling the constraint.

$$
\begin{aligned}
\underset{\substack{a \in \{0,1\}^n \\ z \in \{0,1\}^d}}{\text{minimize}} \quad & \pm \frac{1}{2n} \sum_{i=1}^{n} r_i a_i + \lambda_0 + \lambda_1 \sum_{j=1}^{d} z_j + \frac{\gamma}{n} \sum_{i=1}^{n} \delta_i \\
\text{subject to} \quad & a_i = \sum_{k=1}^{K} \max(x_{ij} z_j) \\
& \forall i : \delta_i = 1 - \max_{j}(1 - M_{ij}) z_j
\end{aligned}
\tag{4}
$$

**Variation on `MINTY`** Note that `MINTY` can be combined with zero imputation, in which case when the disjunctions are learned, the missingness constraint in (3) is disregarded, which is close to the approach proposed by (Wei et al., 2019), although here conjunctions are learned rather than disjunctions.

## A.3. Experiment details

The baselines are trained by the following parameters. The best values for these hyperparameters are chosen based on the validation test set.

`LASSO`: The values of alpha indicating a $\ell_1$ regularization term on weights range within $[0.1, 0.6]$, where increasing this value will make model more conservative. We allow to fit an intercept and set the precompute parameter to *TRUE* to get the precomputed Gram matrix to speed up calculations (Buitinck et al., 2013).

`XGB`: In `XGB` we range the learning rate (eta) between $[0.3, 1.0]$ where the shrinking step size is used in the update to prevent overfitting. After each boosting step, we can directly get the weights of new features, and eta shrinks the feature weights to make the boosting process more conservative. The maximum depth of the trees is set to 10 since increasing this value will make the model more complex and more likely to overfit (Chen & Guestrin, 2016). The hyperparameters lambda represents the $\ell 2$ regularization term on weights and alpha indicates the $\ell 1$ regularization term. Increasing this value will make a model more conservative.

`DT`: For `DT` we set the criterion to measure the quality of a split using the squared error and used 'best' as the strategy to choose the split at each node. The maximum number of features is set to 7 since we have 7 covariates in the data set. A node will be split if this split induces a decrease of the impurity greater than or equal to 0.001. Complexity parameter 'ccp alpha' is used for Minimal Cost-Complexity Pruning where the subtree with the largest cost complexity that is smaller than 0.005 will be chosen (Pedregosa et al., 2011).

`NEUMISS`: For `NEUMISS` models we define the dimension of inputs and outputs of the NeuMiss block (n-features), set the number of layers (Neumann iterations) in the NeuMiss block (depth) and fix the number of hidden layers in the MLP (mlp depth) as well as the width of the MLP (mlp width). If 'None' take the width of the MLP will be the same as $n$ of covariates of a data set (Le Morvan et al., 2020).

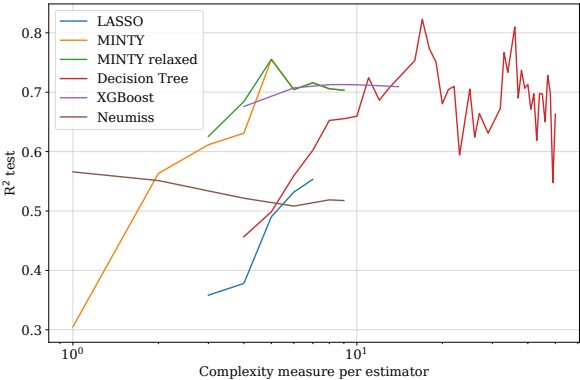

*Figure 1.* Performance on simulated data. The full data set has $n = 2000$ samples and was generated over **10** turns. As a criterion for complexity, we use for MINTY, MINTY$_{relaxed}$ and LASSO the number of non-zero coefficients achieved by regularisation. NEUMISS does not aim at a sparse solution and therefore we give the complexity by the number of covariates used in the model. Note, there might be more parameters to optimize for. The complexity for XGB is defined by the number of estimators used, however, the number of parameters used in total is much larger, and for DT we describe the number of leaves.

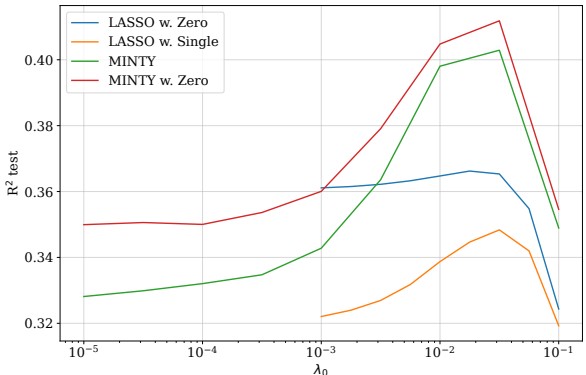

*Figure 2.* Performance on simulated data. The full data set has $n = 200$ samples and was generated over **10** turns.

### A.4. Additional empirical results

In Figure 2 we show preliminary results in our study. We compare predictive performance in terms of the coefficient of determination ($R^2$) on the test set, averaged across multiple draws of synthetic data, against the regularization strength indicated by the $\lambda_0$. We have fit four models namely MINTY, MINTY with zero imputation, and LASSO models with either zero or single imputation by chained equations as baseline models. We find that the MINTY models perform better than the baseline models since they most likely benefit from their nonlinear function class. A potential reason why MINTY with zero-imputation performs better than the MINTY including missing values is that the missingness constraint in (3) is always inactive for MINTY with zero imputation. This leads to the conclusion that MINTY based on zero imputed data has more rules in the model class to choose from. The constraint to improve interpretablity seems to introduce some cost on MINTY when learning rules.

In Table 3 we present the rules learned by MINTY$_{relaxed}$, and we compare a set of learned rules (bottom Table) to the ground truth rules (top Table) from the generated data. We interpret the results by saying that for the first rule, the model learned the correct replacement variables ((Age$\geq$70) OR (Sex= $female$)) and added another (Education$_{low}$) variable to the set, however, the coefficient is low and negative which does not match with the +2 coefficient in the truth model. For the second rule in the top table, MINTY choose (Heart rate$\geq$120BPM) OR (Prior Alzheimer's diagnosis) which does not fully match the true replacement of (Education$_{low}$). The coefficient should be a multiple of the learned on (+1.55 to +3).

*Table 3.* Customized rule sets for predictions based on the ground true rule set $S$ (Top table). And learned rules set and the corresponding coefficients in the bottom table are based on `MINTY`. The results are based on a generated data set with n = 2000 samples and a $p_miss = 0.2$

| TRUE RULES | COEFF. |
|---|---|
| (AGE≥70) OR (SEX= $female$) | +2 |
| (HEART RATE≥120BPM) OR (EDUCATION$_{low}$) | +3 |
| (EDUCATION$_{low}$) OR (PRIOR ALZHEIMER'S DIAGNOSIS) | +2 |
| (AGE≥70) OR (HEART RATE≥120BPM) | -5 |
| INTERCEPT | +0 |

| LEARNED RULES | COEFF. |
|---|---|
| (AGE≥70) OR (SEX= $female$) OR (EDUCATION$_{low}$) | -0.34 |
| (HEART RATE≥120BPM) OR (PRIOR ALZHEIMER'S DIAGNOSIS) | +1.55 |
| (EDUCATION$_{low}$) OR (PRIOR ALZHEIMER'S DIAGNOSIS) | +1.86 |
| (SEX= $female$) OR (HEART RATE≥120BPM) | -4.21 |
| (EDUCATION$_{low}$) | +0.30 |
| (NEVER MARRIED) | +0.23 |
| INTERCEPT | +0.15 |
| **PREDICTION** | **1.94** |

However, the third rule in both tables matches the true and the learned rules including (Heart rate≥120BPM) OR (Prior Alzheimer's diagnosis) with a rounded coefficient of +2. The learned model suggests replacing (Heart rate≥120BPM) with the literal of (Sex= $female$) while the ground truth indicates that (Age≥70) would be correct. The coefficients for this rule are both negative which captures the general influence of the variables,while the learned one is slightly less (-4.21 to -5). The single literals in rule 5 and 6 in the learned table have no counterpart in the true rule set and can be disregarded.

