# OpenReview forum: "Learning replacement variables in interpretable rule-based models"
_ICML.cc/2023/Workshop/IMLH — IMLH 2023 PosterShortPaper_

### Official Review · Reviewer_GaoK · 2023-06-12
**An intreresting method to improve accuracy and interpretability for handling missing variables**

**Rating:** 6
**Confidence:** 3

**Review:**

Summary: This short paper aims to learn the rules of compensating for missing variables under the assumption that the distribution of missing variables at the training and test time is the same. It proposes a linear rule model (named MINTY) and its optimization algorithm to automatically learn rule sets for selecting a replacement when the features set of a sample contain a missing value.

Novelty: I am not quite familiar with this literature, but the idea and method sound novel to me.

Soundness: The extensive empirical results show that MINTY achieves comparable performance to the baseline models when handling missing data. Additionally, MINTY offers an additional advantage by providing interpretable explanations,
- The evaluation of the method is limited to a synthetic dataset. Also, in the experiment, the percentage of missing variables is 20\%, how does the performance change if the percentage increases? Is the hyperparameter selection of $\lambda_0, \lambda$ more difficult?
- If the number of literals/features is large, does it affect the quality of the learned rule sets?
- In the qualitative results of rule sets (section 4.2), including the source and more detailed descriptions of ground truth rules would be useful. It is difficult to understand how to interpret the ground truth rules and their coefficients.

Clarity/Quality of writing: The paper is well-written and easy to follow.
- In Table 1, it would be helpful to highlight the best performance.

Strength:
1. A simple and clear way to provide interpretability while maintaining the performance of the model, when missing variables are present at training and test time. This is relevant to the workshop theme and is an interesting problem in real-world healthcare applications.
2. The empirical results show the effectiveness of this method both quantitatively and qualitatively. A comprehensive study of this model is provided (complexity, baselines, relaxed variants).

Weakness:
1. Lack some ablation studies that examine the impact of varying feature sizes and the percentage of missing variables.
2. The algorithm is only evaluated on a simulated dataset, and the test set is small (200 samples). The contribution would be more significant if the proposed method had been applied to a real-world dataset.

Minor issues:
- The caption of table 2: $p_{miss}$

---

### Official Review · Reviewer_aAi1 · 2023-06-17

**Rating:** 7
**Confidence:** 3

**Review:**

This paper presents a novel approach called MINTY, which addresses the challenge of predicting with missing values in rule-based models. Rule models are known for their interpretability, but standard rule models struggle when confronted with incomplete input data during testing. MINTY introduces the concept of replacement variables to learn compact yet accurate rule models. These replacement variables act as substitutes for each other when one or more variables are missing. By incorporating these replacements into the model, MINTY enables a trade-off between interpretability and goodness of fit while considering missing values at test time.

The authors demonstrate the effectiveness of MINTY through preliminary experiments and compare its predictive performance against baselines. By avoiding direct dependence on imputation or missingness indicators, MINTY offers a principled approach for making predictions with missing values. The use of replacement variables allows for the prediction of missing features based on observed ones, enhancing interpretability without sacrificing accuracy. While further validation and comparison are needed, MINTY presents a promising methodology for learning rule-based models with missing values, with potential applications in domains where interpretability is crucial, such as clinical scoring systems.

---

### Official Review · Reviewer_bdrZ · 2023-06-18
**The reseach topic is interesting, and the overall quality of this short-paper meets satisfactory standards.**

**Rating:** 7
**Confidence:** 4

**Review:**

This manuscript discussed a new rule-based model called MINTY, designed to manage and predict outcomes in scenarios where data may be missing at testing time. The model aims to be both small and interpretable, while delivering high predictive performance, even when the inputs are incomplete. The structure and quality of this manuscript is good. Some suggestions are listed below:

1. The abstract presents the problem and solution well. However, it is not clear what kind of data or situations where this methodology would be particularly useful.
2. In the introduction, it is not clear what "replacement variables" are before mentioning them in the context of rule models.
3. The paper mentions "redundancy in the covariates set", however, it is not clear what a 'covariates set' is.
4. While explaining the MINTY methodology, please also specify how it differs or improves upon existing methods, especially in the context of handling missing values.
5. The disjunction example "(Age > 20) or (Female)" is useful in understanding how MINTY handles missing values, but why this strategy is effective.
6. The justification for your choice of baseline models could be expanded. Why were these models chosen? How do they compare to MINTY?
7. In 'Simulated Data' section, why a Bernoulli distribution was chosen for the sampling of data points?
8. Why was the ADNI database chosen? How does this improve your study?
9. It would be beneficial to discuss more the strategies used for dealing with missingness mechanisms. As the concept of dealing with missingness is central to the work, understanding how MINTY handles different missingness mechanisms is extremely important to its robustness.
10. Please indicate the potential impacts of this model in healthcare decision-making, and the ethical implications of it.
11. In the conclusion, there is no mention of any limitations or challenges encountered during the research process, which would help in establishing a comprehensive narrative around the MINTY model.

---

### Meta-Review · Area_Chair_5kR4 · 2023-06-18

**Recommendation:** Accept (Poster)
**Confidence:** 5

**Metareview:**

This paper aims to explore methods for compensating missing variables, assuming that the distribution of missing variables remains the same during both training and testing. It introduces a linear rule model, called MINTY, along with an optimization algorithm that can automatically learn rule sets for selecting a replacement when a sample's feature set contains a missing value.

The paper was found to be intriguing, well-motivated, and sound by all reviewers. As a result, they unanimously recommended acceptance.

---

### Decision · Program_Chairs · 2023-06-20

Accept (Poster Short Paper)